# Nanostructure ITO and Get More of It. Better Performance at Lower Cost

**DOI:** 10.3390/nano10101974

**Published:** 2020-10-05

**Authors:** Manel López, Juan Luis Frieiro, Miquel Nuez-Martínez, Martí Pedemonte, Francisco Palacio, Francesc Teixidor

**Affiliations:** 1Department of Electronics and Biomedical Engineering, Faculty of Physics, University of Barcelona, Martí i Franquès Street 1, 08028 Barcelona, Spain; jfrieiro@el.ub.edu (J.L.F.); pedemonte96@gmail.com (M.P.); fpalacio@el.ub.edu (F.P.); 2Institute of Materials, ICMAB-CSIC, Campus UAB, 08193 Bellaterra, Spain; mnuez@icmab.es

**Keywords:** nanostructured ITO, biosensors, dependence of growth parameters: temperature, time and composition

## Abstract

In this paper, we investigated how different growth conditions (i.e., temperature, growth time, and composition) allows for trading off cost (i.e., In content) and performance of nanostructured indium tin oxide (ITO) for biosensing applications. Next, we compared the behavior of these functionalized nanostructured surfaces obtained in different growth conditions between each other and with a standard thin film as a reference, observing improvements in effective detection area up to two orders of magnitude. This enhanced the biosensor’s sensitivity, with higher detection level, better accuracy and higher reproducibility. Results show that below 150 °C, the growth of ITO over the substrate forms a homogenous layer without any kind of nanostructuration. In contrast, at temperatures higher than 150 °C, a two-phase temperature-dependent growth was observed. We concluded that (i) nanowire length grows exponentially with temperature (activation energy 356 meV) and leads to optimal conditions in terms of both electroactive surface area and sensitivity at around 300 °C, (ii) longer times of growth than 30 min lead to larger active areas and (iii) the In content in a nanostructured film can be reduced by 10%, obtaining performances equivalent to those found in commercial flat-film ITO electrodes. In summary, this work shows how to produce appropriate materials with optimized cost and performances for different applications in biosensing.

## 1. Introduction

Nanostructured materials have attracted the attention of the scientific community due to their intrinsic characteristics, like emission and signal amplification, being applicable to medicine for drug delivery and tissue engineering, energy applications, nanotechnology, sensing and biosensing, among others [1,2,3]. In particular, in the field of biosensors, nanostructured materials have gained importance since they can provide increased sensitivities, major current levels and detection area, which can be relevant in early-stage detection point-of-care systems [4,5,6]. One of the first nanomaterials used for biosensing was porous silicon, in the early 1990s. The first reference to this material belongs to Bell Labs, but it was Canham who discovered its red and infrared emission that had a positive influence on research on this material [7,8,9]. Later, Dancil et al. [10] used it for highly sensitive biosensing devices. Other nanomaterials for this purpose were carbon nanowires and nanotubes [11], and silicon carbide nanostructures [12], which exhibit a range of fascinating and industrially important properties, among them the stability of interband and defect-related green-to-blue luminescence and good biocompatibility. Other materials like nanostructured Ge and silicon-germanium (SiGe) were aimed at the enhancement of the electronic device performance for semiconductor devices [13]. One of the most studied nanomaterials in the last years has been indium tin oxide (ITO), whose nanostructuration in the form of nanowires was first observed in 2003 by Wang et al. [14]. Among the aforementioned nanostructured materials, ITO presents different advantages among which are highlighted its compatibility with the standard Complementary Metal Oxide Semiconductor (CMOS) silicon technology and, due to its transparency and conductivity properties, the possibility to simultaneously performing electrochemical and optical measurements [15,16].

ITO presents a conductivity near 10^4^ S/cm [17], a band gap of 3.2 eV and transparency in the visible region of around 90% [18,19]. ITO has been exhaustively studied, analyzed and exploited for years in optoelectronics. Its main uses are in fabricating photovoltaic cells [20], sensor modules [21] and transparent electrode materials for both electrochromic cells [22] and liquid-crystal display devices [23]. In addition, it was demonstrated that ITO has been used as an ideal working electrode for different types of biosensors. One example is its use in the electrochemistry of biomolecules and as platform to immobilize different types of immunoreagents [24]. Recently, some companies have started to commercialize optical transparent ITO screen-printed electrodes [25], however they are still using thin-film layers. In the last few years, a nanostructured variation of ITO electrodes has drawn the attention of the scientific community due to the added value of the increased surface-to-volume ratio, improving and increasing its conductivity and transparency in the visible and infrared range.

The use of nanostructured ITO films with improved behavior compared with thin-film layers in many applications has been demonstrated by us and other authors [26]. J.H. Huang et al. reported ITO nanorod films based on the chromophore type electrochromic device. There is a proportionality between active area and device sensitivity. In this sense, it was demonstrated that higher contrast can be achieved with ITO nanowires than with classical ITO thin-film-based sensors for electrochromic molecules subjected to redox reactions [27]. Other authors compared nanostructured ITO films for developing dye-sensitized solar cells with the usual ITO thin films. They detected a better efficiency and short-circuit current than those developed with ITO thin films [14]. Recently, nanostructured ITO has shown some exotic optical properties as broadband, omnidirectional antireflective properties and others appropriate for emerging applications of terahertz (THz) technology. It has been confirmed that this material is a good dichroic mirror for Fourier Transform Infrared Spectroscopy (FTIR) [28].

The purpose of this paper is to provide a better understanding of the physical and structural characteristics of nanostructured ITO to thereby be able to improve the properties of future biosensors based on this material. This will permit the enhancement of the actual commercial ITO electrodes for both performance and trade-off costs. To do this, a systematic study has been carried out, showing some previous results and presenting new experiments explaining the main characteristics of nanostructured ITO, its main properties and their dependence with growth parameters. The understanding of these parameters enables the tuning of our target material, choosing suitable growth conditions. This knowledge will permit to increase sensibility, accuracy and degree of detection of nanostructured ITO based biosensors. This is the first time, to our knowledge, that such deep study including various growth conditions has been done to find the optimal fabrication conditions for ITO electrodes.

We will focus our paper on one of the most widespread techniques of fabrication for this kind of material: electron beam evaporation (EBE) [29]. We chose this deposition process because it provides a high material utilization efficiency, and a deposition rate as low as 1 Å/s, which permits a very precise control on both thermodynamics and geometry of the films. Other techniques described in the literature are chemical vapor deposition (CVD) [30], sputtering [31], spray pyrolysis [32] and atomic layer deposition (ALD) [33]. We have analyzed the structural and electrochemical dependence of the deposited films with temperature, time and Sn composition. To do the characterization we employed scanning electron microscopy (SEM), X-Ray photoelectron spectroscopy (XPS) and energy dispersive X-Ray spectroscopy (EDX). Finally, the characteristics that make this material an exceptional working electrode will be analyzed: Transparency in the visible and near infrared part of the spectrum, conductivity and active area. This analysis was performed by cyclic voltammetry (CV). In Section 1, an introduction to the use of nanostructured materials as well as the main objective of this paper has been presented. Section 2 shows the structural dependence of the nanostructured ITO with temperature, time and relative elements concentration. Section 2 deals with the search of the best growth conditions, presenting new results on the dependence of the growth with temperature and time. Section 3 shows the characterization by cyclic voltammetry, conductivity and transmittance of the ITO samples analyzed in Section 2, focusing the analysis on the increase of its active area and its evolution with temperature and time, obtaining the best conditions from the electrochemical point of view. Section 4 presents the electrochemical characterization of a glycidoxypropyltrimethoxysilane (GOPTS)-functionalized surface, and the comparative behavior analysis of these nanostructured and thin-film ITO functionalized electrodes. Section 5 corresponds to the discussion of the different results and finally we present the conclusions in Section 6.

## 2. Growth of ITO Nanowires. Characteristics and Dependences

### 2.1. Sample Preparation

Different procedures leading to nanostructured ITO using either physical [34,35] or chemical [36] methods have been reported. The most popular methods are sputtering, chemical vapor deposition (CVD) and electron beam evaporation (EBE). We focus our task in EBE because of our earlier experience with this technique. With EBE, we obtain nanostructured ITO avoiding any metal catalyst, additional oxygen atmosphere or control on the crystallographic growth direction. The experimental EBE set up employed consisted of a Pfeiffer vacuum classic 500 chamber equipped with a Ferrotec Genius electron beam controller and a Ferrotec Carrera high-voltage power. Samples were grown on top of (100) p-Si wafers at a base pressure of 10^−6^ mbar with an acceleration voltage of 6 keV and (i) by modifying the temperature from 100 to 500 °C maintaining constant the concentration ratio of In_2_O_3_: SnO_2_ at 90:10 wt.%, (ii) fixing the temperature and the concentration ratio increasing the deposition time, and (iii) by modifying the concentration ratio of In_2_O_3_: SnO_2_ from 60:40 to 90:10 wt.% at a fixed substrate temperature of 300 °C. For all samples the deposition rate was set at 1 Å/s, and evaporation time was ranged from 1 to 30 min.

The samples were analyzed using a JSM-7100F (JEOL Ltd. Tokyo, JP 2003), a field-emission scanning electron microscope (FE-SEM), equipped with an energy dispersive X-ray spectrometer (EDX). This equipment allows detecting either morphological variations as well as changes in the composition of the samples. A LED source of 15 kV permits to achieve resolutions of 0.8 nm.

The analysis of mean nanowire length, mean diameter, number of nanowires per square micrometer and mean distance between nanowires for the different samples was calculated using Gatan Inc. (New York, NY, USA) DigitalMicrograph software (GMS version 3.7.4) (accessed on 2018). GMS is a standard software for analyzing TEM and SEM images. GMS permits to implement or download scripts from its website to customize the analysis. In our case we used D. R. G. Mitchell’s *Measure_Feature.s* script [37,38], which allowed us to calculate the mean radius and length, as well as the standard deviation for the different ITO samples.

### 2.2. Growth Dependence with the Substrate Temperature, Deposition Time and Sn Concentration

When a surface is modified with pseudovertically standing nanostructures, we have a greater surface area. As a first approximation, there is an increase that is proportional to the length of the nanostructure, *z*, the diameter, *2r*, and the number of nanowires, *i*, following the expression (Equation (1)):(1)$$A\;=\;A_{0}+\underset{i}{\sum }2\pi \cdot r\cdot z\cdot i$$
where A is the final surface area and A_0_ the initial one, before the addition of the nanostructures. By incorporating such nanostructures into sensors and bionsensors we can provide an increase in their active surface area without affecting its overall dimension (Equation (1)). This growth reaches a maximum area that depends on the substrate temperature, the deposition time and partial concentration of the different elements. All these parameters modify the nanowire’s density and dimensions, the final morphology of the sample, and the surface and interfacial surface tension between sample and analyte used. Once the maximum surface value has been reached, the increase in the density and size of the nanowires implies a decrease in the active area [34,35].

ITO samples grown at temperatures below 125 °C form a thin and continuous film layer, whereas, at temperatures higher than 200 °C, the deposited samples grow forming ITO nanowires. Figure 1 demonstrates this behavior. Sample 1a was grown at 100 °C and the SEM image shows no nanostructuration, just a homogeneous thin-film layer. On the contrary, sample 1b was grown at 500 °C. In this case, ITO nanowires are observed. Both samples were grown by EBE at a rate of 1 Å/s and a target of In_2_O_3_: SnO_2_ at 90:10 wt.% concentration. The nanostructuration of the ITO samples were observed by SEM. The 80,000 × amplified images of the nanostructures for each substrate temperature are shown in Appendix A in the electronic Appendix A. Nanowires can be observed from 200 °C to 500 °C. Appendix A also shows the EDS spectra for those samples grown at 300 °C and 500 °C, appreciating the same composition for both samples. Physical properties were presented in Appendix A in the Appendix A. The Powder X-ray Difraction (PXRD) data indicate that, in the absence of other more complete studies, the ITO generated is amorphous, that is to say that the nanowires are in a metastable state. The transmittance observed for all the samples (except that grown at 200 °C) is over the 80% for the as-deposited samples and close to the 100% when annealing is performed.

Similar results were obtained by Kumar et al. [35], who grew ITO nanowires also by EBE, by Fung et al. [39], who produced ITO nanowires by dc sputtering, and by Johnson et al. [40] using CVD. Authors that have reported the fabrication of ITO nanowires have described two mechanisms for growing such nanostructures: vapor–liquid–solid (VLS) and self-catalytic VLS [41,42,43]. The first one requires relatively high growth temperatures, normally ranged between 700–1000 °C. The second one needs relatively low growth temperatures: from 250 to 600 °C [44,45,46]. Within the frame of the self—catalytic mechanism, the binary phase diagram of the In–Sn system [4,47] presents a eutectic point located near 125 °C. Some studies [18,44,48,49,50,51] have demonstrated that, at substrate temperatures near 100 °C, In and Sn oxides are in the solid state, meaning that no In_2_O_3_:SnO_2_ seeds are formed because the temperature is below the melting points of In/Sn and thus the eutectic point of the alloy is not reached. This noncatalytic activity of the material compound results in the absence of nanowire growth. At temperatures above the eutectic point it is observed the formation of nanowires with identical composition to the base components used in the growth process. All the nanowires have the catalyst particle at their tips, which can be observed in Figure 1b and has been confirmed to be made of the same material as the nanowire by micrograph techniques [35], thus supporting the self-catalytic VLS hypothesis [41,42,43].

In [34], the self-catalytic VLS was interpreted with the Stranski–Krastanov method of epitaxial growth [52] where ITO first grew up forming thin-film monolayers. This growth continues until a physicochemical feature triggers a change, giving rise to the catalyst particles due to the temperature and enhancing the growth of the nanowires. The nanowire growth and density depend on three parameters: temperature, time and deposition rate, temperature being the most relevant factor.

#### 2.2.1. Growth Dependence with Substrate Temperature

Temperature is the main growth factor in ITO nanowires. Their growth is performed by bonding of atoms evaporated from the target and deposited on the substrate. It is well known that the difference between the chemical potential of the In_2_O_3_:SnO_2_ deposited over the flat surface is less than the one over the ITO nanowires. This fact permits the nanowire’s growth, either in diameter or length. The diffusion coefficient of dissolved atoms *D* allows the increase of the dimensions of the nanowires. This diffusion can be adjusted by Equation (2):(2)$$D=\;D_{0}e^{-E_{D}/kT}$$
where *E_D_* is the activation energy for the atomic diffusion of the dissolved atoms. The higher the substrate temperature is, the higher the density, and larger the length and diameter, as can be observed in Table 1 and Figure 2. Different samples grown at different temperatures were analyzed by SEM to study the evolution of ITO nanowires. Figure 2a was performed using the scipy.optimize library, which permitted us to adjust the measurements provided by GMS to an exponential function using least squares approximation (*scipy.optimize.curve_fit()*). Figure 2b was obtained using the *LinearRegression* library, which belongs to the *sklearn.linear_model* python package.

The results presented in Table 1 and Figure 2 evidences how, at a certain activation energy (~0.356 eV), by increasing the temperature, it increases the mean nanowire length and diameter, decreases the neighboring distance and increases the number of nanowires per area unit.

#### 2.2.2. Growth Dependence with Deposition Time

To analyze more in detail the growth of ITO nanowires, we extended the deposition time up to 30 min at a fixed temperature of 300 °C. Table 2 shows the evolution of length and diameter with the exposure time. Plotting these results (Figure 3) and applying a linear regression, we determine a growth rate of 17.46 nm/min. In a previous work [34] it was stated that the growth of the nanowires was explained by the Stransky–Krastanov growth model. This is well corroborated by our actual data, as the intercept value result of the linear regression is –83.9 nm. This means that the nanowires do not grow up instantly, but after an ITO thin layer is initially formed to allow that the nanowires grow on it. Figure 3 shows the evolution of mean length as well as mean diameter with the deposition time. From the data we can also extrapolate that nanowires start to grow (with the temperature and rate conditions employed in this work) after ~5 min of deposition with a mean diameter of ~17 nm, which increases as time goes by. The ITO nanowire growth, as for other type of nanowires occurs via crystallization from the liquid droplet. The diameter of the nanowire is correlated with the diameter of the liquid droplet on the surface. A theoretical growth model described by Schmidt et al. [53,54] describes how the balance of forces associated to surface effects makes that the z axes will be the predominant address, but the equilibrium balance of forces makes that the diameter also increases with the deposition time.

Similar results were obtained by Kumar et al. in [35] but for higher deposition rates. However, it is clear that in both cases the nanowire’s length and density increase with the growth duration. Kumar et al. [35] have shown that the length of the nanowires varied from 500 nm to 5 µm as the growth duration changed from 8 to 30 min.

#### 2.2.3. Growth Dependence with SnO_2_ Concentration

One of the growth parameters that is least varied in ITO is the relative ratio of In and Sn. It is common to find the optimal growth concentrations of ITO as In_2_O_3_ 90 wt.%–SnO_2_ 10 wt.%. However, such a high concentration of In affects the price of the sensor, since the scarcity of this oxide in nature is well known, as well as the high cost required to obtain it. Indeed, a Sn doping level at this ratio is optimal in terms of carrier density, with a value between 10^21^–2 × 10^21^ cm^−3^ [55], at least an order of magnitude above bare In_2_O_3_. The level of SnO_2_ doping modulates the carrier density in the crystal structure and hence the electrical and optical properties. Theoretical and experimental dependence of carrier density, transmittance, grain size and band gap energy with SnO_2_ doping level was observed by different authors [56]. Below 8% SnO_2_ doping, the carrier concentration decreases due to a distortion of the lattice structure by SnO_2_ [57]. Above 20% SnO_2_ doping, the probability of two or more Sn atoms occupying adjacent cation positions increases (this means that part of the defective donor levels are being compensated), diminishing the conducting properties of the crystal structure [58].

However, and as previously discussed, it is well known how expensive and difficult is to find In in nature. The price of In in the market oscillates between $390 and $475 per kg. As a comparison, the price of Sn in the same market oscillates between $16 and $19 per kg. An increase in the percentage of Sn would imply a reduction in the price of future devices. For this reason, a set of samples of different In_2_O_3_:SnO_2_ ratios (60–40, 70–30, 80–20 and the well-known reference 90–10 in wt.%) were fabricated and analyzed by SEM. The objective of this experiment was to find a range of SnO_2_ concentration in which the growth of ITO nanowires behaved similarly to that described above for the 90–10 wt.% reference. Growth temperature, time and deposition rates were 300 °C, 30 min and 1 Å/s, respectively. Figure 3 shows two different scenarios. For SnO_2_ ratios above 30 wt.%, no nanowires are observed. In these samples, only an ITO rough layer is seen, their composition is the nominally expected and confirmed by EDX results. For SnO_2_ compositions below 30 wt.% an increasing high load of nanowires is detected. Figure 4 shows how the ITO surface evolves with increasing In_2_O_3_:SnO_2_ ratios, staring from 60/40. The surface roughness increases with the concentration until it gets up to the formation of nanowires for concentrations below 30 wt.% of SnO_2_. For a SnO_2_ concentration of 20 in wt.%, ITO nanowires are visible, increasing their density until reaching a maximum at 10 wt.%. An in-depth analysis that electrochemically determines how the surface behaves based on all these parameters is necessary to obtain the best characteristics of sensors based on these types of ITO.

## 3. Characterization by Cyclic Voltammetry, Conductivity and Transmittance of ITO Nanowires at Different Growth Parameters

The electrochemical measurements were done with a commercial potentiostat SP150 (Biologic, France). The software used to carry out both measurements and adjustments was EC-Lab V11.12 (accessed on March 2016). Measurements were made on a three-electrode polycarbonate cell, with a geometrically projected area for the working electrode (WE) of 6 mm diameter, resulting in an area of around 0.28 cm^2^. ITO samples were used as WE, a platinum wire of 0.5 mm diameter was used as a counter electrode (CE), and KCl saturated Ag/AgCl was used as a reference electrode (RE). The electrochemical experiments based on cyclic voltammetry (CV) were all done in aqueous solution (ultrapure water, Milli-Q Millipore, MERCK, Burlington, Massachussetts, USA). As a redox couple, 5 mM potassium hexacyanoferrate Fe(II) and Fe(III) couples, [Fe(CN)_6_]^−3/−4^, were used. CV is very useful to obtain the active area of the active surface area of the sensor. To determine the conductivity of the samples a B2912A dual precision source-measurement unit (SMU) with a minimum source and measurement resolution of 10 fA/100 nV was used following a four-point probe system. To assure that there was no appreciable Schottky barrier, different samples grown similarly were measured with different shapes without detecting any appreciable variation in the sheet resistance. The behavior of the Faradaic current peaks related with oxidation and reduction process is described by the Randles–Sevcik Equation (3) [58,59,60]:(3)If=knFACnFνDRT
where *k* = 0.4463 is a nondimensional proportionality constant; *n* corresponds to those electrons involved in the redox process (1 in the case of [Fe(CN)_6_]^−3/−4^; *F* is the Faraday’s constant (96,485 C mol^−1^) and R is the universal gas constant (8.314 J mol^−1^ K^−1^); *A* is the active area of the electrode, in cm^2^; *ν* is the rate at which the potential is swept, in V/s; *D* is the analyte’s diffusion coefficient in cm^2^ s^−1^; *C* is the analyte’s concentration in mol cm^−3^ and *T* is the solution temperature in K. The active area can be calculated doing successive CVs at different scan rates and plotting the maximum signal corresponding to the oxidation current against the square root of the scan rate. The slope of this plot is directly proportional to the electroactive surface area of the electrode.

The sheet resistance of the samples was measured in bulk, without applying any additional pressure on the material and preserving the integrity of the ITO nanowires as much as possible. The measurement was done with a four-point probe system based on a Keysight Technologies B2912A (Keisight Technologies, Santa Rosa, California, USA) dual precision source-measurement unit (SMU), with a minimum source and a measurement resolution of 10 fA/100 nV. The conductivity of the employed electrode does not influence the intensity of the oxidation–reduction peak but rather its position. Indeed, the more resistive the electrode, the wider the separation between reduction and oxidation peaks. This is due to the reversibility of the redox reaction taking place when employing a nonideal conductive electrode, compared to its ideal counterpart which would facilitate the reversibility.

To measure the transmittance of the samples we used a Bentham PVE300 Spectral Response (Bentham Instruments, Reading, UK) analyzer coupled to an integrating sphere. A monochromator was used with a grid resolution between 0.3 nm and 0.6 nm and dispersion between 2.7 nm and 5.4 nm selected wavelengths from two light sources: a 75 W xenon light source for wavelengths between 300 and 700 nm and a 100 W quartz halogen light source for wavelengths between 700 nm and 1700 nm. Appendix A (Appendix A) shows the transmittance spectra for ITO nanowires as a function of the deposition temperature for a constant concentration ratio of In2O3:SnO2 (90:10 wt.%) and deposition time (30 min). No noticeable changes were observed in the transmittance spectra from what we reported in some earlier works [14,61] and those reported in the literature [35].

### 3.1. Dependence of the Surface Area, Conductivity and Transmitance with Substrate Temperature

The samples’ growth at different substrate temperatures were studied using cyclic voltammetry and conductivity. These measurements permit to determine the area, transmittance and impedance differences associated with this parameter. Figure 5 shows the results obtained by cyclic voltammetry, using the aforementioned redox couple.

The maximum and minimum values from Figure 5 correspond to the oxidation and reduction peaks of the cyclic voltammetry. From these values it is possible to represent the oxidation and reduction current as a function of the scan rate. Figure 6 shows the linear dependence of both the oxidation and reduction current peak’s height with the square root of the voltage scan rate, well defined by the Randles–Sevcik equation (Equation (3)) after removing the capacitive current by EC-Lab software (accessed on March 2016).

As can be observed from Equation (3) the slope of Figure 6 is directly proportional to the active area. In this sense, Figure 6 presents both oxidation and reduction current peaks for the different scan rates. Low scan rates present higher linearity because of processes arising at the electrode solution became as quasistatic and quasireversible. Table 3 outlines the results of the different linear regressions. As previously commented, the area of the electrochemical cell is ~0.28cm^2^. Therefore, those samples that show the formation of ITO nanowires should present an increase of the active area. This evidence is observed for all the samples except for the electrode prepared at 200 °C, where nanostructuration is not enough to show that increase. On the other hand, the sample grown at 300 °C presents both the highest available electroactive surface area and lowest sheet resistance [34,35].

For all the samples, transmittance as a function of growth temperature was also analyzed. The results show a behavior of around 80% for as-deposited samples. These results are also presented in Table 3. After annealing in a rapid thermal processing unit (60 min at 600 °C), the transmittance spectra increased to values around 100% in the visible wavelength range for all the samples [34,58,61,62].

Measurements in sheet resistance show a cutback when increasing the deposition temperatures, reaching a minimum located around 300 °C, which could be explained by a change in the final solid phase of the system. Table 3 summarizes the results obtained for cyclic voltammetry, transmittance and conductivity for ITO nanowires grown at different substrate temperatures. Samples prepared at 300 °C behave optimal for redox reactions to reach the electrode surface. For lower temperatures, nanostructuration is not enough to provide an increased surface area. On the other hand, for higher temperatures, there is a higher density of nanowires that hinders the electron exchange between the solution and the electrode. Kumar et al. [35] presented similar results that were correlated with the atomic percentage of tin in the samples. The difference between the sheet resistance values presented in Table 3 and those reported by Kumar et al. [35] (which are around 1 kΩ sq^−1^) can be explained by differences in the experimental growth process, local differences in the geometry of the nanowires and the thickness of the deposit.

### 3.2. Dependence of the Surface Area, Conductivity and Transmitance with Deposition Time

The samples growth with different deposition time were also studied using cyclic voltammetry and conductivity following the aforementioned methodology. These measurements permit to determine the effective area. No substantial changes were detected in the transmittance value or the conductivity of nanostructured ITO as a function of the deposition time. The temperature selected to do this analysis was 300 °C. This temperature provided the best results in effective area and conductivity compared with the other substrate temperatures. Different growth times, ranging from 5 to 30 min, were used to analyze the electrochemical behavior of the samples. All the samples were studied by cyclic voltammetry with voltage scan rates ranging from 10 to 700 mV sec^−1^. The higher the scan rate, the higher the current signal response linked to the redox reaction. As it was previously stated, it is possible to perform a linear regression of either the oxidation or reduction current vs the square root of the scan rate. The slope of this linear regression is directly proportional to the active area, as can be observed in Equation (3) (Figure 7).

Figure 8 shows the evolution of the active area as a function of deposition time. These points were adjusted using the *curve-fit* function provided by the python Scipy library. The best fit corresponds to a logarithmic evolution of the active area as a function of time. From this curve it is possible to obtain a cost–benefit ratio in terms of the active area ratio and deposition time. In this sense and as an example, doubling the deposition time, going from 30 min to 1 h would correspond to an increase in active area of only 12%. Experimentally, deposition times ranging from 30 to 60 min are enough to achieve a significant increase in the active area without an excessively high cost.

### 3.3. Dependence of the Surface Area on Sn Concentration

The evolution of the nanostructured ITO surface active area with the Sn concentration was again carried out using cyclic voltammetry. The samples studied were grown at 300 °C for 30 min, varying the Sn concentration from 40 wt.% and decreasing to the well-known 10 wt.%.

The results from Table 4 are clear. The most sensitive electrodes are nanostructured ITO samples with the well-known 90/10 concentration in wt.%. The larger is the ratio of tin, the lower the efficiency of the nanostructured material. The nanostructuration of the material and therefore the increase of the active area, with regard to the thin-film reference, continues even with ratios of Sn equal or probably higher than 40 wt.% of tin. One of the most interesting results shown in Table 4 is that nanostructured ITO 80/20 and 70/30 wt.% electrodes present higher sensitive area than the commercial ITO 90/10 thin-film electrodes with a much lower price. As expected, when the In concentration is lowered so it is the cost of the electrode, the material lowers sensitivity but nonetheless the material is far superior to the commercial ITO 90/10 thin-film electrodes. Thus, nanostructuration modulates the behavior of the as made electrode permitting that a nanostructured ITO 70/30 wt.% has a better behavior than commercial thin-film 90/10 wt.% ITO electrodes, as it is observed in Table 4. Depending on the sensing material, it is possible to reach a balance that minimizes the cost and provides the necessary sensitivity for the detection of the substance under scrutiny.

## 4. Electrochemical Characterization of the Functionalized Surface. Analysis of the Behavior of the Nanostructured vs Thin-Film ITO Working Electrodes

The increase of the electroactive surface area was demonstrated by different authors attaching an organic molecule to the previously functionalized surface [63,64]. This kind of functionalization is widely used as a first step for the immobilization of biomolecules on oxidized surfaces for the fabrication of on-chip devices. For this study, a glycidoxy compound containing a reactive epoxy group was chosen, the so-called 3-glycidoxypropyltrimethoxysilane (GOPTS). The attaching process was extensively explained in [26]. The binding was performed between GOPTS and the ITO surface by the immersion of the nanostructured sample in a solution of GOPTS at 4% (*v*/*v*) in toluene. To detect the link between the ITO electrode (nanostructured or not) and the electroactive surface, the complete system was analyzed with XPS. XPS measurements were centered on both intensity and binding energy of O1s and Si2p bonds for determining the behavior of these peaks for nanostructured ITO electrodes with and without GOPTS, respectively (Figure 9). Table 5 presents and summarizes these results, showing an overall increase of the signal intensity on the GOPTS coated ITO compared with the nonfunctionalized electrodes.

O1s bond reveals the first sign of ITO functionalization. A shift in the peak splitting due to Si-O bond formation was measured and presented in Table 5. It is correlated with GOPTS attached to the ITO surface and explained by the covalent binding of -OH groups on the ITO surface. For the non-GOPTS-coated ITO, O1s peaks were fitted to Gaussian distributions centered at 530 and 531.2 eV, meanwhile for the functionalized samples peaks shift to 530.1 and 532.4 eV.

Si2p binding energy, centered at 102 eV, shows a weak peak for the nonfunctionalized samples, probably associated to impurities of the sample. However, for the GOPTS-functionalized ITO the peak appears magnified five times, indicative of the presence of Si on the substrate surface, as a result of GOPTS functionalization. This peak position agrees with the information provided by Martene et al. [65] in their review on organosilane technology in coating applications.

The presence of the epoxy fragment on the ITO surface was also detected by CV for both thin-film and nanostructured ITO electrodes. CV also allows us to analyze the level of improvement introduced by the use of a nanostructured ITO electrode. In this case, CV was carried out and the results are shown in Figure 10. It is important to remark the effect of the GOPTS binding into the ITO surface. The inset of Figure 10 shows how the current signal decrease for that ITO sample bound to GOPTS compared with the nonfunctionalized ITO sample. The measurements were done using an electrolyte of 5 mM [Fe(CN)_6_]^−3^/[Fe(CN)_6_]^−4^. The current decrease is explained by the increase of the impedance due to the presence of GOPTS in the surface of the ITO samples. Figure 10 (1 and 2) also shows the improvement in signal intensity performed by the nanostructuration. To avoid parasite effects that could be associated to the [Fe(CN)_6_]^−3,−4^, samples were immersed in a 0.5 mM solution of 6-(ferrocenyl)hexanethiol (FHT) in N,N-dimethylformamide to bind the FHT to the complex GOPTS-ITO. The CV was then performed in 10 mM NaCl electrolyte and the FHT was responsible for the electronic exchange in the oxidation reduction process. CV oxidation and reduction peak height showed an increase of around 400% for the nanostructured sample compared with the ITO thin-film one. This important evolution is due to the higher electroactive surface area of the nanostructured ITO compared with thin-film ITO and correlates with the results obtained in paragraphs 2 and 3 (Figure 10). Future analysis will be done, tuning the nanostructured ITO with the growth parameters, to perform biosensors for specific analytes.

## 5. Discussion

In this paper, the different growth conditions that allow increasing the efficiency of ITO as a substrate electrode for the development of biosensors have been analyzed in detail. These growth conditions allow us to determine the best deposition temperature, compensate both the cost and the manufacturing time and adjust the type of electrode according to need. This work is mainly focused on the evolution of ITO nanowires as a function of deposition temperature, time and concentration. The main concern of this nanostructuration is to enlarge the active area without affecting the overall dimension. A larger electroactive surface area implies a marked increase in the intensity current when redox reactions related with electrochemical detection processes, take place.

It was demonstrated that the main factor of the ITO nanostructuration is temperature, this being the key factor to be considered for obtaining high-quality nanostructured ITO working electrodes. It was also demonstrated that there is a threshold activation temperature, near 150 °C, from which nanowires begin to grow exponentially. This growth can be correlated with the minimum temperature at which ITO nanostructuration starts and corresponds with the eutectic point of the SnO_2_–In_2_O_3_ system, promoted by a self-catalytic VLS growth. Moreover, from the electrochemical study of the samples, it was found that 300 °C is the optimum temperature for growing nanostructured ITO. This growth temperature maximizes the electrochemical area, improves the electrical conductivity and maintains a high optical transmittance. The main consequences behind this experimental result have to do not only with the increase in surface area, but also with the optimization of the surface, which could allow a greater use of the sensor surface. When the growth temperature is very high the density of nanowires is also very high. This high density of nanowires generates capillary forces whose side effects are opposed to the desired performance of the electrodes. At the end, a very high-density nanostructured sample will just behave as a planar surface. Our results seem to demonstrate that the trade-off is found at 300 °C, enough nanostructuration to increase the detection, enough low density to avoid such capillary forces.

The evolution of the nanostructure growth with time permits to adjust and analyze more deeply its behavior. The experimental results presented in Section 2 confirm that there is a linear growth of both length and diameter of the nanowires with the deposition time. It allows the beginning of the process to be predicted, and thus controlled. The linear regression, with a coefficient of determination above 90%, presents a negative intercept value, meaning that nanowires growth requires of a different ITO evolution prior to their formation. This process defines two phases: an initial one where there is a pseudoepitaxial growth of an ITO thin film over the substrate, to continue with a growth based on the self-catalytic formation of the nanowires. This behavior perfectly corresponds with the Stranski–Krastanov growth model, as we predicted in a previous paper [34]. However, a larger size of the nanowires does not have to correspond to a linear increase in the active detection area. The experimental results indicate that the evolution of the active area with the deposition time is adjusted to a logarithmic function. These results show that, after an initial growth period practically linear, the subsequent evolution seems to tend asymptotically to a limit value. In this sense, a significant increase in time does not mean a significant increase in the active area. Deposition times between 30 min and one hour are more than enough to achieve a significant increase in the active surface for commercial electrodes.

Finally, the evolution of the different samples with composition shows that the best one corresponds to the well-known In_2_O_3_ 90 wt.%–SnO_2_ 10 wt.%. The reason of this good behavior is because of the level of Sn doping enhances the carrier density in the crystal structure and therefore enhances its electrical properties. A Sn doping concentration around 10% (atomic weight) is optimal in terms of carrier density, with a value between 10^21^–2×10^21^ cm^−3^ [65]. For Sn concentrations higher than 20%, the probability of two or more Sn atoms occupying adjacent cation positions increases, reducing the conducting properties of the crystal structure [66]. However, the high cost of In is well known, so a decrease in its concentration can reduce the cost of the sensor for those devices that do not require such a high sensitivity. The price of one kg of In is around $400. However, the same quantity of Sn has a cost of $17. Samples based on In_2_O_3_ 70–80 wt.%–SnO_2_ 30–20 wt.% with nanostructuration display performances comparable or superior to the commercial common concentration (90 wt.% vs 10 wt.%), but at a much lower cost. The best compromise solution is to have a nanostructured ITO material, giving the same or similar performances at a lower cost. Growing nanostructured ITO makes their sensitivity higher to that of current devices grown as thin films, making this proposed concentration a good candidate to manufacture low-cost devices that maintain a high reliability.

In any case, it seems clear that nanostructured growth makes ITO working electrodes have a better response and increase their sensitivity, as shown in Section 4. Nowadays ITO working electrodes are common electrodes and are already integrated in some devices [24], so a simple modification in the manufacturing process could introduce important improvements in such devices.

## 6. Conclusions

In this paper we have performed an in-depth structural and electrochemical study to provide a better understanding of the characteristics of nanostructured ITO. This knowledge permits to tune the growth of nanostructured ITO to develop better biosensors based on this material. From the ITO growth dependence with temperature we have demonstrated that it exists an activation temperature, of about 150 °C, from which nanowires begin to grow exponentially. This growth can be correlated with the minimum temperature from which ITO nanostructuration starts and corresponds with the eutectic point of the SnO_2_–In_2_O_3_ system, promoted by a self-catalytic VLS growth. The dependence with time shows a linear evolution both in length and diameter. The linear regression indicates that the nanowires’ growth process requires two phases: one in which a small ITO surface layer is grown to continue with the growth of the nanowires. From the electrochemical study of the samples, it was found that 300 °C was the optimal temperature for nanostructured ITO. Lower temperatures mean lower active areas, but higher temperatures imply a higher density and the occurrence of other superficial forces that diminish the active area. The evolution of the nanostructured sensing area with time behaves logarithmically so it is not necessary to spend more than 30 or 60 min to get a device with very good sensing conditions. We have also studied the dependence of ITO with Sn concentration, showing ITO nanostructuration up to 30 wt.% and active area higher than the commercial ITO thin-film electrodes. This means that is possible to reduce the price of the electrodes and increase the sensitivity by introducing nanostructured conditions and modifying Sn concentration.

Overall, the results presented will improve the properties of future biosensors based on this material. The knowledge hereby presented will enable the tuning of our target material, choosing suitable growth conditions that will permit us to enhance sensibility, accuracy and degree of detection of nanostructured ITO-based biosensors.

## Figures and Tables

**Figure 1 nanomaterials-10-01974-f001:**
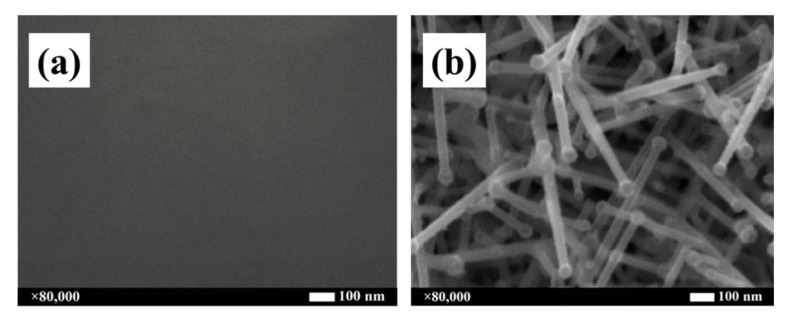
SEM images of indium tin oxide (ITO) thin layers grown for 30 min at 1 Å/s and (**a**) 100 °C or (**b**) 500 °C of substrate temperature.

**Figure 2 nanomaterials-10-01974-f002:**
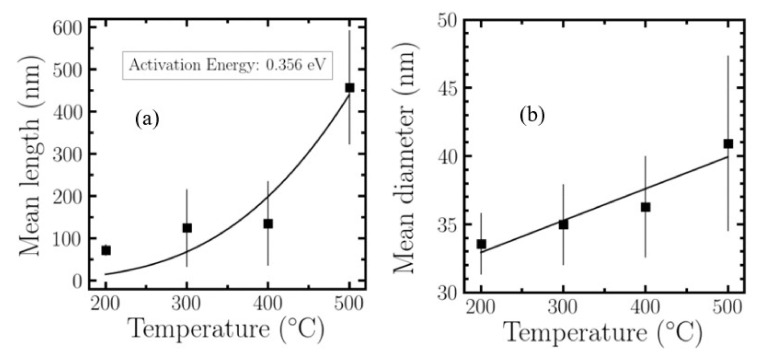
Evolution of the ITO nanowires length (**a**) and diameter (**b**) as a function of the substrate temperature.

**Figure 3 nanomaterials-10-01974-f003:**
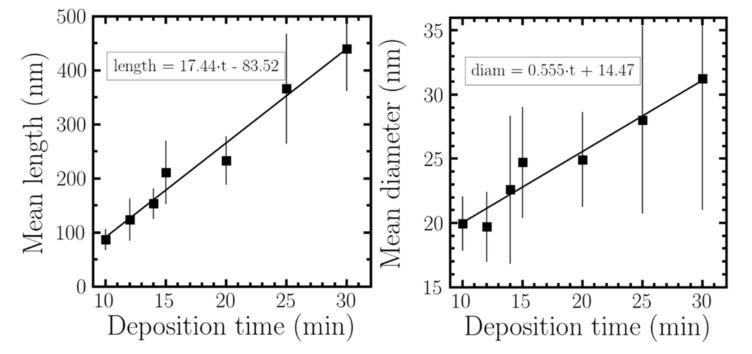
Evolution of the ITO nanowires length and diameter as a function of the deposition time. Straight line shows the different fits for those samples.

**Figure 4 nanomaterials-10-01974-f004:**
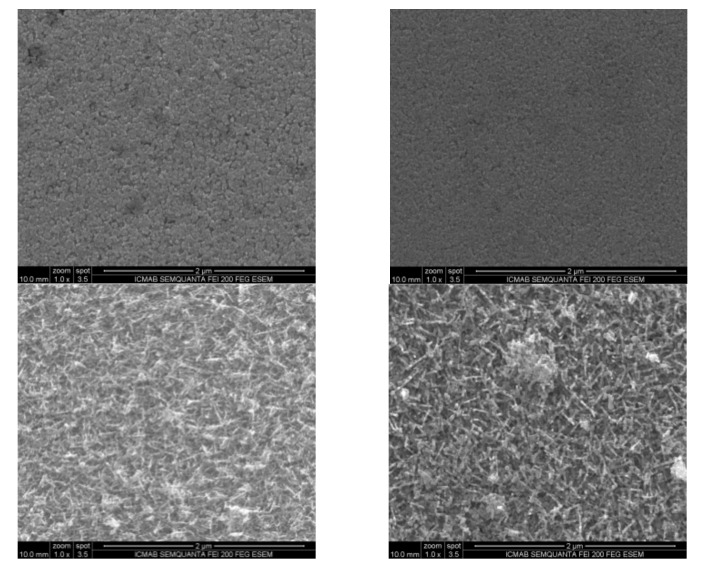
SEM images of ITO deposited at 300 °C, 30 min for In_2_O_3_ concentration of 60 (up left), 70 (up right), 80 (bottom left) and 90 wt.% (bottom right).

**Figure 5 nanomaterials-10-01974-f005:**
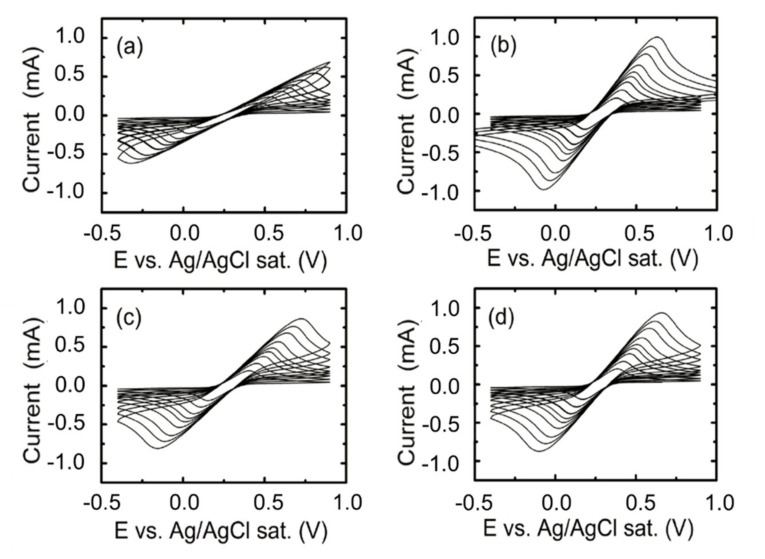
Faradaic cyclic voltammetry of ITO electrodes prepared at (**a**) 200 °C, (**b**) 300°C, (**c**) 400 °C and (**d**) 500 °C. The cycles were taken at scan rates of 10, 25, 50, 75, 150, 250, 350, 500 mV s^−1^ (increasing order in the figures).

**Figure 6 nanomaterials-10-01974-f006:**
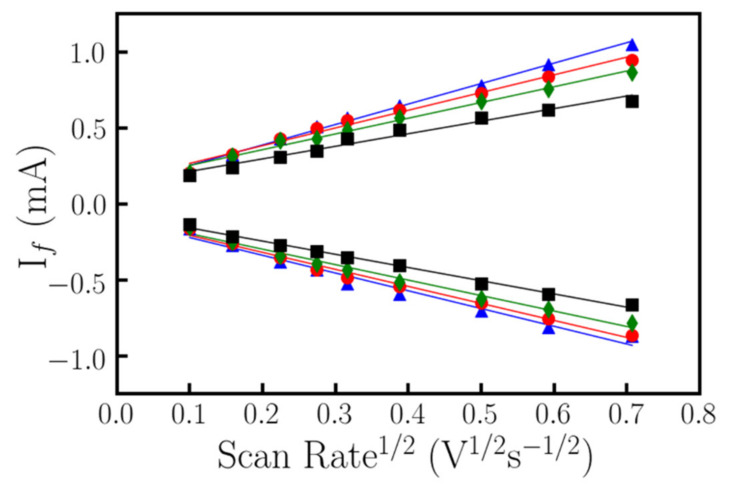
Evolution of the anodic (upper datasets) and cathodic (lower datasets) current peaks versus the square root of the scan rate for ITO electrodes grown at 200 (black squares), 300 (blue triangles), 400 (green diamonds) and 500 °C (red circles) [34].

**Figure 7 nanomaterials-10-01974-f007:**
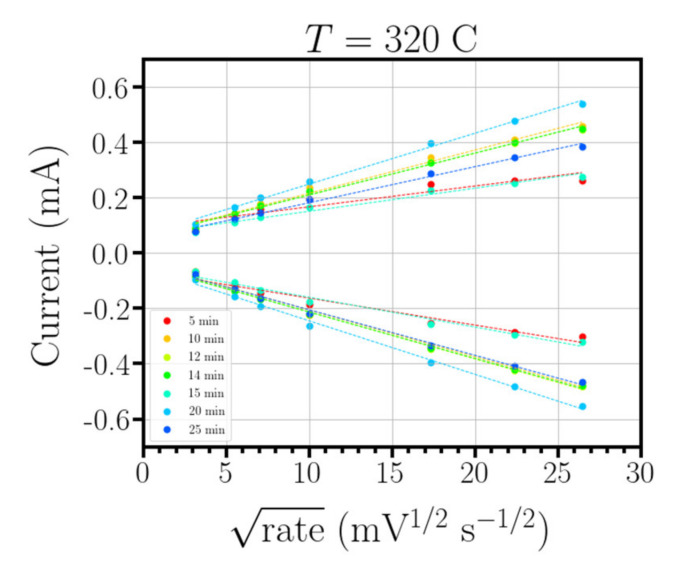
Linear regression of the evolution of the oxidation (positive curve) and reduction (negative curve) current versus the square root of the scan rate. Samples studied range from 5 to 30 min with a step of 5 min.

**Figure 8 nanomaterials-10-01974-f008:**
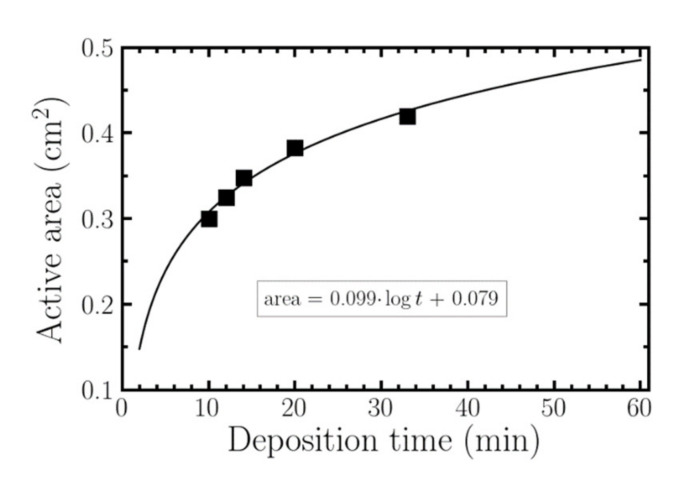
Evolution of the active area as a function of the deposition time. Black squares correspond with the experimental results. The solid line corresponds with the logarithm fit.

**Figure 9 nanomaterials-10-01974-f009:**
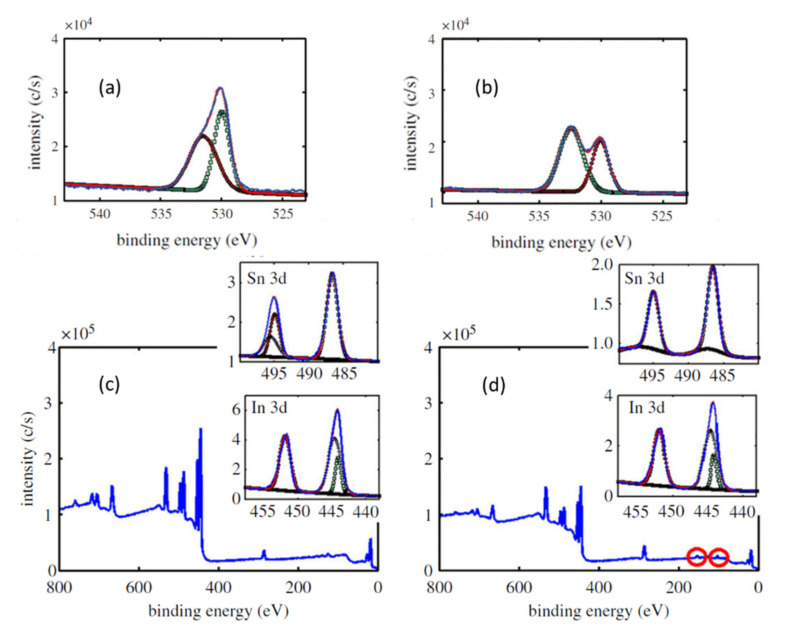
XPS spectra for nanostructured ITO. (**a**) XPS O1s peak spectra for as-deposited ITO. (**b**) XPS spectra for GOPTS-functionalized ITO. XPS wide spectra corresponding to (**c**) as-deposited ITO and (**d**) GOPTS-functionalized ITO.

**Figure 10 nanomaterials-10-01974-f010:**
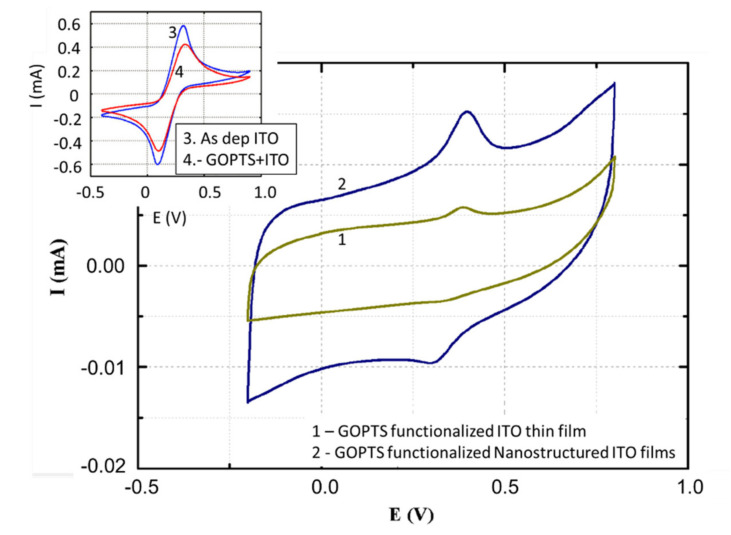
Comparative image of the cyclic voltammetry evolution for the (1) GOPTS-functionalized ITO thin layer and (2) GOPTS-functionalized of nanostructured ITO layers. The inset shows the cyclic voltammetry (CV) difference between as-deposited ITO and GOPTS-functionalized ITO thin-film layers.

**Table 1 nanomaterials-10-01974-t001:** Dependence of nanowire size and density vs substrate temperature.

Temperature (°C)	100	200	300	400	500
**Mean Nanowire Diameter (nm)**	No NW	33.56	34.98	36.29	40.91
**STDV Diameter (nm)**		2.266	2.955	3.736	6.439
**Mean Length (nm)**	No NW	72.23	124.6	135	457.95
**STDV Length (nm)**		13.13	21.95	32.01	135.8
**Number of Nanowires/Area Unit**	-	433 nw/µm^2^	610 nw/µm^2^	625 nw/µm^2^	616 nw/µm^2^
**Nanowire Area vs. Total Area**	-	77.7%	95.4%	97.8%	96.66%
**Mean Distance Between Nanowires (nm)**	-	0.24	In contact	In contact	In contact

**Table 2 nanomaterials-10-01974-t002:** Dependence of nanowire size and density vs time growth.

Exposed Time (min)	Mean Length (nm)	STDV Length (nm)	Mean Diameter (nm)	STDV Diameter (nm)
10	86.98	19.62	19.95	2.139
12	123.7	39.08	19.7	2.735
14	153	28.19	22.59	5.783
15	211	58.72	24.71	4.331
20	233.1	44.59	24.95	3.715
25	365.7	101.3	28.03	7.322
30	439.4	77.5	31.23	10.231

**Table 3 nanomaterials-10-01974-t003:** Evolution of the electroactive area, transmittance and sheet resistance for nanostructured ITO samples grown at temperatures ranged from 200 to 500 °C.

Sample (°C)	Area (cm^2^)	Transmittance (%) @600 nm	Sheet Resistance (Ω/sq)
200	0.31	79	215
300	0.42	82	105
400	0.38	81	155
500	0.40	82	145

**Table 4 nanomaterials-10-01974-t004:** Surface active area values obtained for ITO at different compositions at substrate temperature of 300 °C. A thin-film sample growth at 90/10 wt.% and 100 °C was added as a reference.

Composition In/Sn wt.%	Active Area (cm^2^)
90/10	0.623 ± 0.017
80/20	0.273 ± 0.012
70/30	0.1 ± 0.012
60/40	Not detected
Thin film (reference)	0.029 ± 0.004

**Table 5 nanomaterials-10-01974-t005:** Summary of atomic concentration (%) and Gaussian fit characteristics of O1s and Si2p peaks for bare ITO and glycidoxypropyltrimethoxysilane (GOPTS)-functionalized ITO samples.

	O1s Peak (%)	Si2p Peak (%)	O Peak1 Position (eV)	O Std. Dev.1 (eV)	O Peak2 Position (eV)	O Std. Dev. 2 (eV)	Si Peak Position (eV)	Si Std. Dev. (eV)
Bare ITO	51	0.5	531.2	1.70	530	0.84	101.3	1
GOPTS Func. ITO	58.1	7	532.4	1.35	530.1	0.97	102.4	1.21

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
