# Peer review of "Nanostructure ITO and Get More of It. Better Performance at Lower Cost"

_nanomaterials, 2020, doi:10.3390/nano10101974_

Round 1

Reviewer 1 Report

This paper presents a careful investigation of the growth of nanostructured Indium Tin Oxide by electron beam evaporation. The scientific results are interesting, and the commercial implications of producing effective material with reduced Indium content are significant.
In general the presentation is good, but there are a few points which could be made clearer.
1. Line 183 implies that Table I and Figure 2 give evidence for the quoted activation energy. It is not obvious how the lines in Figure 2 are determined, or how they relate to the diffusion described by Equation 2.
2. From lines 201-202 and 161-163 it appears that the autocatalytic particles nucleate at a diameter of about 17nm and then increase in diameter as the nanowires increase in diameter: it would be interesting to have some comment on this.
3. Table II shows deposition times up to 30 minutes, but Figure 3 only goes up to 25 minutes. Is there some reason for this?
4. In Figure 5(a) the range of the graph cuts off some of the peaks in the voltammetry curves.
5. It would be helpful to have best straight-line fits in Figure 6.
6. Is there any physical explanation of the logarithmic growth of active area with time, as described in Line 343 and Figure 8? From Equation 1 and Figure 3 the total area appears to grow faster than linearly with time. Also, in Figure 8, why do the deposition times appear to be different from those in Table II?
7. It would be easier to assess the significance of the chemical changes if error estimates were included in Table IV. Also, whereas Table IV has 531.2eV the text in Line 391 has 531.5eV.
Minor points:
line 91: he -> the
line 127: Subsrat -> Substrate
Table I: It would be better to use nm throughout, and not introduce Angstroms.
line 352: Might be better to include the word "active".

Overall, this is a significant paper which deserves to be published.

Reviewer 2 Report

This manuscript reported the fabrication and application of ITO nanowires (NWs). However, the data and discussion provided were insufficient. In addition, the experimental process is unclear. The readers cannot obtain significant information from this study. Based on the viewpoints listed below, I cannot recommend this manuscript to be published.

Comment:

  1. The morphologic evolution of the different growth temperature and deposition time of ITO NWs should be provided.
  2. The absorption spectra of all ITO NWs from UV to near IR range should be supplied, because no spectra were found in this draft.
  3. How to measure the conductivity of all ITO NW? The measurement procedure is important because the sample is a nanowire.
  4. The crystalline information, such as XRD and TEM, should be supplied and discussed.
  5. The composition of ITO NWs, such as EDS and XPS, should be provided and discussed. Because I cannot find the EDS and XPS data, only the numbers in Table.
  6. In Figure 9, the number 1 and 4 are the same GOPTS functionalized ITO thin film layers, but the CV curves are different. In addition, the discussion in this section is unclear.
  7. The authors should carefully compare their results with the literatures.

Reviewer 3 Report

This is an excellent study on the properties of ITO nanowires synthesized at different temperature. The authors discuss the optimal condition from both their physical properties and economic cost of their fabrication. The experimental data included are vast and complete, with appropriate interpretations.

Except for lacking XPS profiles and some typographic errors, this manuscript can be accepted as it is. Please check once again for the typographic errors.

l.91: of he → of the

l.186: Table I number of nanowires   nw/um2 →  nw/μw2

l.386: Prior to Table IV, XPS profiles should be presented.

Round 2

Reviewer 2 Report

No further comment.